# MODIS-Derived Estimation of Soil Respiration within Five Cold Temperate Coniferous Forest Sites in the Eastern Loess Plateau, China

**Junxia Yan [1,\*], Xue Zhang [1], Ju Liu [2], Hongjian Li [1] and Guangwei Ding [3]**

[1]   Institute of Loess Plateau, Shanxi University, Taiyuan Shanxi 030006, China; zhangx0229@163.com (X.Z.); hongli@sxu.edu.cn (H.L.)
[2]   Shanxi Academy of Forestry, Taiyuan Shanxi 030006, China; liuju821107@163.com
[3]   Chemistry Department, Northern State University, Aberdeen, SD 57401, USA; Guangwei.Ding@northern.edu
\*   Correspondence: yjx422@sxu.edu.cn; Tel.: +86-351-7010700

**Abstract:** Soil respiration ($R_s$) is seldom analyzed using remotely sensed data because satellite technology has difficulty monitoring various respiratory processes in the soil. We investigated the potential of remote sensing data products to estimate $R_s$, including land surface temperature (LST) and spectral vegetation indices from the Moderate Resolution Imaging Spectroradiometer (MODIS), using a nine-year (2007–2015) field measurement dataset of $R_s$ and soil temperature ($T_s$) at five forest sites at the eastern Loess Plateau, China. The results indicate that soil temperature is the primary factor influencing the seasonal variation of $R_s$ at the five sites. The accuracy of the model based on the observed data is not significantly different from the model based on MODIS-derived nighttime LST values. There was a significant difference with the model based on MODIS-derived daytime LST values. Therefore, nighttime LST was the optimum LST for estimation of $R_s$. The normalized difference vegetation index (NDVI) consistently exhibited a stronger correlation with $R_s$ when compared to the green edge chlorophyll index and enhanced vegetation index. Further analysis showed that adding the NDVI into the model considering only $T_s$ or nighttime LST could significantly improve the simulation accuracy of $R_s$. The models depending on nighttime LST and NDVI showed comparable accuracy with the models based on the in situ $T_s$ and NDVI. These results suggest that models based entirely on remote sensing data from MODIS have the potential to estimate $R_s$ at the cold temperate coniferous forest sites. The performance of the model in other vegetation types or regions has also been proved. Our conclusions further confirmed that it is feasible for large-scale estimates of $R_s$ by means of MODIS data in temperate coniferous forest ecosystems.

**Keywords:** soil respiration; soil temperature; land surface temperature; vegetation indices; MODIS data; cold temperate coniferous forests

## 1. Introduction

Soil respiration ($R_s$) is the second largest carbon flux between terrestrial ecosystems and the atmosphere [1]. Consequently, small changes in $R_s$ will have a large impact on atmospheric $CO_2$ concentration and climate warming. Therefore, an accurate estimation of the spatial–temporal variation in $R_s$ is required to assess the carbon budgets of terrestrial ecosystems [2] and to understand the effect of global warming on $R_s$ [3,4].

Since $R_s$ is a combined flux from plant roots and microorganisms from different soil depths [5], several factors and their interactions affect $R_s$ rates. Soil temperature ($T_s$) and soil moisture ($W_s$) are considered to be the most important factors controlling the $CO_2$ flux [6,7]. In addition, other factors, such as vegetation types [8,9], composition and quantity of litter [10], soil organic carbon [11,12],

soil nitrogen [13], and fine root biomass [13], also impact $R_s$. Many semi-empirical models have been used for predicting the spatial and temporal variability of $R_s$ using in situ measurement data, including $T_s$, $W_s$, and vegetation characteristics [1,14–16]. However, on a large spatial scale, these factors are difficult to obtain with in situ measurements because of their distinct spatial and temporal changes [13,15]. Due to spatial data products providing us a broad range of spatial coverage and regular temporal sampling, we speculate that if the spatial data relating to soil temperature and moisture from satellite remote sensing can be used in an $R_s$ model, then $CO_2$ efflux over large spatial and temporal scales can be estimated [3].

Satellite techniques have been used for the estimation of the spatial distribution of gross primary productivity (GPP), net primary productivity (NPP), and net ecosystem exchange (NEE) [17]. However, $R_s$ estimations based on remote sensing products remain problematic because it is difficult for remote sensing to monitor various respiratory processes in the soil [3,18]. Previous studies have reported that remote sensing data could be used to establish $R_s$ models. For example, the land surface temperature (LST) night-driven model can simulate the temporal variation of $R_s$ in deciduous and evergreen forest sites [19]. In a study of $R_s$ of forest landscapes in Saskatchewan, Canada, Wu et al. [20] showed that an accurate estimation of $R_s$ could be inferred with the product of the normalized difference vegetation index (NDVI) and the nighttime LST derived from the Moderate Resolution Imaging Spectroradiometer (MODIS) imagery as the independent variable in regression equations. Furthermore, the accuracy of $R_s$ models based only on remotely sensed data are comparable with those based on in situ measured data [3]. However, the performance of satellite-driven $R_s$ models in other vegetation types or regions, such as a semiarid region, like the Loess Plateau, China, requires further study [18,19].

In this study, we evaluated the potential to estimate $R_s$ using remote sensing data products. We used our nine-year dataset of field measured $R_s$, $T_s$, and $W_s$ on five forest sites at the middle of Lvlian Mountain in the eastern Loess Plateau of China and the MODIS product dataset (http://ladsweb.nascom.nasa.gov/data/search.html) corresponding to the sites to analyze the correlations between $R_s$ and LST values and in situ measured $T_s$, and subsequently determined the optimum temperature predictors. Then, we investigated the correlations between $R_s$ and the three vegetation indexes (VI, e.g., the normalized difference vegetation index (NDVI), the green edge chlorophyll index ($CI_{green\ edge}$) and the enhanced vegetation index (EVI)), and selected the best VI predictors. Finally, we built empirical models of $R_s$ from the remotely sensed data, using different statistical approaches based on the optimum temperature and vegetation index for each site, and we also evaluated the accuracy of these $R_s$ models.

## 2. Materials and Methods

### 2.1. Study Sites

The study site is located in the Pangquangou National Natural Reserve in the Guangdi Mountains of Shanxi Province, China. The region has a temperate continental monsoon climate. Between 1977 and 2011, the annual average precipitation was 604.9 ± 177.5 mm, ranging from 935.0 mm in 1967 to 358.4 mm in 1997, and 60% of the precipitation occurred mostly in the summer months based on a provincial rainfall station near the site. The annual average temperature was 4.3 °C, and mean temperatures in January and July were −10.2 and 17.5 °C, respectively. The altitude of the area ranges from 1400 to 2700 m. The dominant trees in the region are composed of *Larix principis-rupprechtii Mayr.* (Prince Rupprecht's Larch), *Picea wilsonii Mast.* (Wilson Spruce), *Picea meyeri Rehd. et Wils.* (Meyer Spruce) and *Populus davidiana Dode* (Wild Poplar), which is mostly located between 1700 and 2600 m above sea level, accounting for 60% of the area. The forests in the area have not been cut or thinned since 2000, which are in a state of natural growth. The experiment was carried out in five forest sites, including an evergreen needle-leaf forest (ENF), an evergreen and deciduous needle-leaf mixed forest (NMF), and three deciduous needle-leaf forests (DNF-1, DNF-2, and DNF-3), which are located at

different altitudes ranging from 1790 to 2387 m. The physical and chemical characteristics of the sites are listed in Table 1.

**Table 1.** Summary of characteristics for all five sites.

| Sites | NMF | ENF | DNF-1 | DNF-2 | DNF-3 |
|---|---|---|---|---|---|
| Latitude | N 37°53′08.4″ | N 37°52′34.4″ | N 37°53′33.7″ | N 37°53′24.3″ | N 37°53′03.4″ |
| Longitude | E 111°25′56.6″ | E 111°26′31.0″ | E 111°31′05.0″ | E 111°30′15.1″ | E 111°30′34.5″ |
| Elevation (m) | 2163 | 1986 | 2387 | 2264 | 2105 |
| Slope (°) | ~16 | ~8 | ~25 | ~32 | ~1 |
| Aspect | SW | SW | SW | SW | SW |
| Soil texture | Loamy sand | Loamy sand | Loamy sand | Sandy loam | Sandy loam |
| Soil depth (cm) | 10–35 | 10–30 | 10–35 | 10–30 | 10–30 |
| SBD (g cm$^{-3}$) [a] | 0.73 | 1.26 | 1.04 | 1.11 | 1.27 |
| WHC (%) [b] | 37.25 | 20.32 | 30.62 | 24.19 | 27.47 |
| Plant combination | Coniferous mixed forest | Evergreen coniferous forest | Deciduous coniferous forest | Deciduous coniferous forest | Deciduous coniferous forest |
| Dominant species | *Picea wilsonii Mast.* (Wilson Spruce), *Larix principis-rupprechtii Mayr.* (Prince Ruprecht's Larch) | *Picea wilsonii Mast.* (Wilson Spruce) | *Larix principis-rupprechtii Mayr.* (Prince Rupprecht's Larch) | *Larix principis-rupprechtii Mayr.* (Prince Rupprecht's Larch) | *Larix principis-rupprechtii Mayr.* (Prince Rupprecht's Larch) |
| Stand density (tree ha$^{-1}$) | 950 | 675 | 1175 | 1025 | 925 |
| DBH (cm) [c] | 22.9 ± 8.7 | 29.6 ± 9.0 | 18.7 ± 8.2 | 26.6 ± 11.1 | 28.1 ± 10.3 |

[a] Soil bulk density; [b] Water holding capacity; [c] Diameter at breast height.

## 2.2. Soil Respiration Measurement

The $R_s$ was measured at the five sites using an LI-COR 6400 portable photosynthesis system (LI-COR, Environmental Division, Lincoln, NE, USA) connected to a standard soil chamber (6400-09). In each site, nine or more PVC chambers, which were made of polyvinyl chloride pipe, were permanently installed with a 2-m spacing between them before measuring $R_s$. The aboveground living plants were removed, and the litter was left in the chamber. All measurements were performed during the day from 10:00 AM to 14:00 PM. The $R_s$ measurement process and the equipment calibration were described in Li et al. [7]. At the 10-cm depth ($T_{10}$) near the chamber, the soil temperature was measured by a thermocouple probe (6400-13, LI-COR, Environmental Division, Lincoln, NE, USA) simultaneously with the $R_s$ measurement. After the initial measurements, we continuously observed the soil temperature at 5- ($T_5$) and 15-cm ($T_{15}$) depths. The $W_s$ values from the 0- to 10-cm soil depth near the chamber were measured by an oven drying method at 105 °C. The leaf area index (LAI) was measured simultaneously with $R_s$ measurements by using LAI-2200C (LI-COR, Environmental Division, Lincoln, NE, USA) only in 2015. The soil bulk density (SBD) at 0–10, 10–20, and 20–30 cm was measured using the volumetric core method. The measurements of SBD and stand density were made within three 10 m × 10 m areas. These measurements were made monthly, during the growing season, from July 2007 to October 2015, and a total of 58 measurements were recorded at each site.

## 2.3. MODIS Land Surface Products

We used three land surface MODIS products for our analysis. They were downloaded from NASA's Earth Observing System Data and Information System (http://ladsweb.nascom.nasa.gov/data/search.html). We used the Terra MODIS 8-day surface reflectance data (MOD09A1, 500 m), and the Terra and Aqua MODIS 8-day LST (MOD11A2 and MYD11A2, 1 km). Table 2 shows three spectral vegetation indices (VI) calculated from the surface reflectance product of MOD09A1. The nine LST temperature types were used in the analysis. $LST_{td}$ and $LST_{tn}$ are the LST at Terra's 10:30 AM/PM overpasses, respectively. $LST_{ad}$ and $LST_{an}$ are the LST at Aqua's 1:30 AM/PM overpasses, respectively. $LST_{tav}$ is the mean of $LST_{td}$ and $LST_{tn}$, and $LST_{aav}$ is the mean of $LST_{ad}$ and $LST_{an}$. $LST_{av}$ is the mean of $LST_{td}$, $LST_{tn}$, $LST_{ad}$, and $LST_{an}$. $LST_{dayav}$ is the mean of $LST_{ad}$ and $LST_{td}$. $LST_{nightav}$ is the mean of $LST_{an}$ and $LST_{tn}$.

Pixels containing each study site from the MODIS land surface products (MOD09A1, MOD11A2, and MYD11A2) were extracted for data analysis by using five study sites' geo-location information (latitude and longitude). The values of VI and LST of the $R_s$ measurement days for each site were obtained from the two consecutive 8-day composites by linear interpretation.

**Table 2.** Vegetation indices calculated from MODIS 8-day surface reflectance product.

| Vegetation Index | Formulation | Reference |
|---|---|---|
| Normalized Difference Vegetation Index | $\text{NDVI} = \frac{p_{nir} - p_{red}}{p_{nir} + p_{red}}$ | [21] |
| Enhanced Vegetation Index | $\text{EVI} = 2.5 \times \frac{p_{nir} - p_{red}}{p_{nir} + 1 + 6.0 \times p_{red} - 7.5 \times p_{blue}}$ | [22] |
| Green Edge Chlorophyll Index | $\text{CI}_{\text{green edge}} = \frac{p_{nir}}{p_{green}} - 1$ | [23] |

$p_{green}$, $p_{blue}$, $p_{red}$, and $p_{nir}$ are reflectance of the green, blue, red, and near-infrared (NIR) band in the MOD09A1 product, respectively.

### 2.4. Data Processing and Analysis

#### 2.4.1. Methods for $R_s$ Modelling

Previous studies have established that $T_s$, $W_s$, and vegetation productivity are the three most important abiotic and biotic factors influencing $R_s$ [6,7]. However, due to Pearson's Product Moment correlation coefficient between $R_s$ and $W_s$ at all sites not being statistically significant (Figure S1), we only selected the independent variables of $T_s$, LST, and VI as proxy indicators to build the $R_s$ model.

The exponential and Arrhenius-type functions (Equations (1) and (2)), in which the measured $R_s$ is the dependent variable and temperature, including the measured $T_s$ and MODIS LST data, is the independent variable, were used to explore the correlations between $R_s$ and the measured $T_s$ at three depths (e.g., $T_5$, $T_{10}$, and $T_{15}$), as well as $R_s$ and the MODIS LST values at different passing times. Based on the coefficient of determination ($R^2$) and root mean square error (RMSE) of each of the fitted equations, we determined the best temperature predictors for further analysis:

$$R_s = R_{ref} \times e^{Q \times T}, \tag{1}$$

$$R_s = R_{ref} \times e^{E_0 \times \left( \frac{1}{T_{ref} - T_0} - \frac{1}{T - T_0} \right)}, \tag{2}$$

where $R_s$ is the measured $R_s$ ($\mu\text{mol CO}_2 \text{ m}^{-2} \text{ s}^{-1}$), and $T$ refers to the measured $T_s$ or LST (°C). $Q$ (°C$^{-1}$) represents the rate of $R_s$ change with respect to temperature. $E_0$ (K) is the activation energy parameter that represents the $R_s$ sensitivity to $T$. $T_0$ is the lower temperature limit for the $R_s$, which is fixed at 227.13 K (−46.02 °C), similar to the original model of Lloyd and Taylor [24]. $T_{ref}$ is the reference temperature and it is set to 283.15 K (10 °C). $R_{ref}$ ($\mu\text{mol CO}_2 \text{ m}^{-2} \text{ s}^{-1}$) in Equation (1) represents $R_s$ when $T$ is 0 °C, whereas $R_{ref}$ ($\mu\text{mol CO}_2 \text{ m}^{-2} \text{ s}^{-1}$) in Equation (2) represents the $R_s$ at $T_{ref}$.

Next, we analyzed the correlations between $R_s$ and VI values (NDVI, EVI, and CI$_{\text{green edge}}$) with linear functions and exponential functions, respectively (Equations (3) and (4)). Based on the $R^2$ and RMSE, the best VI predictors were selected for further analysis:

$$R_s = a + b \times VI, \tag{3}$$

$$R_s = a \times e^{b \times VI}, \tag{4}$$

where a and b are the fitted parameters, and VI is one of the three vegetation indexes.

Based on above analysis, the following equations (Equations (5)–(10)) were used to build six 2-variable models of $R_s$ with $T$ and VI. The independent variables were chosen from the best-fitted equations of Equations (1)–(4) that had the highest $R^2$ and lowest RMSE:

$$R_s = a + b \times T \times VI, \tag{5}$$

$$R_s = a + b \times T + c \times VI, \tag{6}$$

$$R_s = a \times e^{(b \times T + c \times VI)}, \tag{7}$$

$$R_s = a \times e^{(b \times T)} \times VI^c, \tag{8}$$

$$R_s = R_{ref} \times e^{E_0 \times \left(\frac{1}{T_{ref}-T_0} - \frac{1}{T-T_0}\right) + c \times VI}, \tag{9}$$

$$R_s = R_{ref} \times e^{E_0 \times \left(\frac{1}{T_{ref}-T_0} - \frac{1}{T-T_0}\right)} \times VI^c, \tag{10}$$

where a, b, and c are the fitted parameters, which differ depending on the model. $T$ (°C) and VI are the corresponding optimal independent variable $T_s$ or MODIS LST, and VI. $T_{ref}$ is the reference temperature, which was set to 283.15 K (10 °C). $R_{ref}$ (µmol $CO_2$ m$^{-2}$ s$^{-1}$) represents the soil respiration at $T_{ref}$.

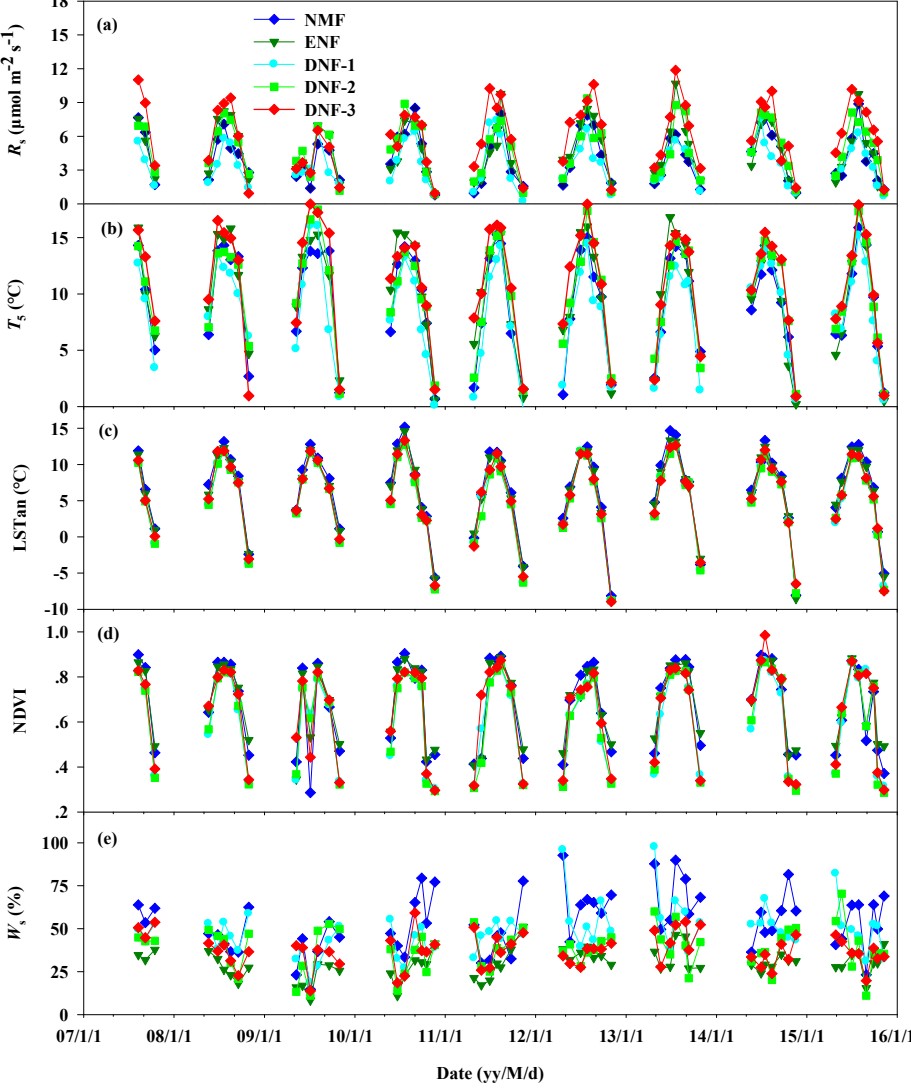

**Figure 1.** Seasonal variations in the (**a**) soil respiration rate ($R_s$, µmol $CO_2$ m$^{-2}$ s$^{-1}$), (**b**) soil temperature at 5 cm depth ($T_5$, °C), (**c**) land surface nighttime temperature from MODIS-Aqua (LST$_{an}$, °C), (**d**) normalized difference vegetation index (NDVI), and (**e**) soil water content at 0 to 10 cm ($W_s$, %) at the five sites during measurement.

2.4.2. Statistical Analysis

All statistical analysis was conducted using SPSS 17.0 (SPSS Inc., Chicago, IL, USA). All plots were drawn using SigmaPlot 11.0 (Systat Software Inc., Chicago, IL, USA). The mean $R_s$ of each site and all chambers was used for statistical analysis. One-way ANOVA was applied to compare the mean differences of $R_s$ for the relevant biotic and abiotic factors between the five study sites. A linear regression model between the MODIS LST and in situ measured $T_s$ was used to confirm the feasibility of using MODIS LST in estimating $R_s$. One-way ANOVA, based on the $R^2$ and RMSE values from each site, was also examined to compare the goodness of fit to the models driven by the in situ measured $T_s$ with that of the models solely considering LST, and post hoc procedure of Duncan was used to determine differences between sites or groups. Akaike's information criterion (AIC) was used to compare the goodness of fit to the models between the single and double variable model. The models were validated using the method of training/evaluation and splitting cross-validation [25]. The model performance was evaluated by statistical indicators, which included the $R^2$, RMSE, and model utilization efficiency (EF) of the estimated residuals.

## 3. Results

*3.1. Seasonal Variations of $R_s$*

Similar to the seasonal variations of $T_s$ and NDVI, $R_s$ showed an obvious seasonal pattern during the study period (Figure 1). The maximum $R_s$ usually appeared at the mid-growing season and corresponded to the maximum $T_s$ and NDVI values, except for one in July of 2009, which corresponded to the lowest $W_s$ recorded during the whole experiment period (Figure 1). One-way ANOVA result illustrated that among the five sites, DNF-3 exhibited the highest $R_s$, DNF-1 showed the least $R_s$, and the difference of the average $R_s$ varied among the sites (Table 3).

$T_5$ and $LST_{an}$ exhibited a similar seasonal trend (Figure 1), with a maximum in summer and minimum at the start and end of the growing season. $T_5$ at DNF-1 was significantly lower ($p < 0.05$) than at DNF-3, but this was not significantly different ($p > 0.05$) than at sites NMF, ENF, and DNF-2. However, $LST_{ad}$ and $LST_{an}$ were not characterized with a significant difference ($p > 0.05$) among the five sites (Table 3). Additional analysis indicated that the correlations between the $T_s$ values measured at different depths and all of the LST values were all highly significant (Table 4). Furthermore, the correlations between $T_s$ with nighttime LSTs ($LST_{tn}$ and $LST_{an}$) were significantly stronger than that between $T_s$ with daytime LSTs ($LST_{ad}$ and $LST_{td}$), indicating that for an $R_s$ estimation, the nighttime LST values were better than the daytime LST values. Additionally, among the three depth $T_s$ values, $T_5$ had the best correlation between the $T_s$ values and LST values.

$W_s$ at the five sites showed a large temporal fluctuation with the occurrence of precipitation events during the measurement. $W_s$ was above 50% of the water holding capacity (WHC) except for in July 2009 (Figure 1). There was a significant difference observed in the average $W_s$ among the sites except for between DNF-2 and DNF-3 ($p < 0.05$). Among the five sites, the NDVI values did not have a significant difference ($p > 0.05$), as they ranged from 0.62 ± 0.20 to 0.68 ± 0.19, with a seasonal coefficient of variation (CV) of 24.35 through 34.00% (Table 3). The maximum NDVI typically occurred in the mid-growing period, except for one occasion in July of 2009 (Figure 1), which corresponded exactly to the least $W_s$ (Figure 1).

**Table 3.** Mean and coefficient of variation (CV, %) of soil respiration ($R_s$, μmol $CO_2$ m$^{-2}$ s$^{-1}$), soil temperature at the 5-cm depth ($T_5$, °C), land surface temperature (LST$_{ad}$ and LST$_{an}$, °C), normalized difference vegetation index (NDVI), and soil water content at 0 to 10 cm ($W_s$, %) at the five sites during measurement.

| Site Code | $R_s$ | | $T_5$ | | LST$_{ad}$ | | LST$_{an}$ | | NDVI | | $W_s$ | |
|---|---|---|---|---|---|---|---|---|---|---|---|---|
| | Mean | CV | Mean | CV | Mean | CV | Mean | CV | Mean | CV | Mean | CV |
| NMF | 4.24 ± 2.27 ab | 53.62 | 9.33 ± 4.71 ab | 50.49 | 15.21 ± 4.85 a | 31.92 | 6.71 ± 5.89 a | 87.77 | 0.68 ± 0.19 a | 28.34 | 54.19 ± 17.20 e | 31.74 |
| ENF | 4.76 ± 2.54 b | 53.48 | 10.18 ± 4.96 ab | 48.72 | 15.08 ± 4.72 a | 31.27 | 6.27 ± 5.73 a | 91.41 | 0.69 ± 0.17 a | 24.35 | 29.62 ± 8.32 a | 28.09 |
| DNF-1 | 3.57 ± 1.94 a | 54.36 | 8.45 ± 4.61 a | 54.62 | 14.55 ± 4.99 a | 34.29 | 5.30 ± 5.69 a | 107.24 | 0.62 ± 0.20 a | 32.88 | 48.57 ± 14.26 c | 29.37 |
| DNF-2 | 4.95 ± 2.39 b | 48.33 | 10.20 ± 4.63 ab | 45.38 | 14.54 ± 4.98 a | 34.26 | 5.22 ± 5.74 a | 109.83 | 0.62 ± 0.21 a | 34.00 | 38.29 ± 12.31 b | 32.13 |
| DNF-3 | 6.11 ± 2.94 c | 48.20 | 11.08 ± 5.03 b | 45.44 | 15.13 ± 4.73 a | 31.26 | 5.67 ± 5.72 a | 100.82 | 0.65 ± 0.21 a | 31.91 | 37.50 ± 9.50 b | 25.34 |
| All | 4.73 ± 2.57 | 54.15 | 9.85 ± 4.84 | 49.14 | 14.90 ± 4.83 | 31.79 | 5.84 ± 5.74 | 97.77 | 0.65 ± 0.20 | 31.75 | 41.64 ± 15.35 | 38.97 |

Data are means ± standard deviations ($n = 58$). Values in the same column followed by the different letters are significantly different ($p < 0.05$) based on the Duncan test.

**Table 4.** Pearson correlation coefficients (r) among four land surface temperatures (LST) and in situ measured temperatures ($T_s$) at the five sites during the measurement.

| Temperature | NMF | | | ENF | | | DNF-1 | | | DNF-2 | | | DNF-3 | | |
|---|---|---|---|---|---|---|---|---|---|---|---|---|---|---|---|
| | $T_5$ | $T_{10}$ | $T_{15}$ | $T_5$ | $T_{10}$ | $T_{15}$ | $T_5$ | $T_{10}$ | $T_{15}$ | $T_5$ | $T_{10}$ | $T_{15}$ | $T_5$ | $T_{10}$ | $T_{15}$ |
| LST$_{an}$ | 0.88 | 0.86 | 0.82 | 0.92 | 0.90 | 0.87 | 0.90 | 0.90 | 0.85 | 0.91 | 0.89 | 0.85 | 0.92 | 0.90 | 0.89 |
| LST$_{tn}$ | 0.89 | 0.87 | 0.83 | 0.91 | 0.89 | 0.87 | 0.89 | 0.89 | 0.83 | 0.89 | 0.87 | 0.83 | 0.91 | 0.89 | 0.88 |
| LST$_{td}$ | 0.73 | 0.69 | 0.62 | 0.83 | 0.79 | 0.74 | 0.73 | 0.70 | 0.61 | 0.72 | 0.68 | 0.62 | 0.77 | 0.73 | 0.69 |
| LST$_{ad}$ | 0.68 | 0.64 | 0.58 | 0.74 | 0.71 | 0.65 | 0.72 | 0.68 | 0.59 | 0.69 | 0.65 | 0.59 | 0.74 | 0.70 | 0.65 |

$T_5$, $T_{10}$, and $T_{15}$ is soil temperature (°C) at a depth of 5, 10, and 15 cm, respectively. LST$_{td}$ and LST$_{tn}$ is the daytime and nighttime land surface temperature observed by the Moderate Resolution Imaging Spectroradiometer (MODIS) onboard Terra satellites, respectively. LST$_{ad}$ and LST$_{an}$ is the daytime and nighttime land surface temperature observed by MODIS onboard Aqua satellites, respectively. The correlation was all significant at the 0.01 level ($n = 58$).

### 3.2. Correlations between $R_s$ and $T_s$ and LST

The correlations between $R_s$ and the temperatures, including three $T_s$ values and nine MODIS LST values, were all highly significant for each site (Table 5, Figure 2), indicating that both $T_s$ and LST values could be used to predict $R_s$. Furthermore, with the $R^2$ and RMSE values of the fitted equations, the $T_5$ equation was the best one using $R_s$ with $T_s$ at the five sites, and $LST_{an}$ was the best using LST (Table 5, Table S1).

**Table 5.** Results of the statistical analysis of the fitted Equations (1) and (2) relating soil respiration to temperatures on the five sites for all measured data during the measurement.

| Model | Temperature | NMF | | ENF | | DNF-1 | | DNF-2 | | DNF-3 | |
|---|---|---|---|---|---|---|---|---|---|---|---|
| | | $R^2$ | RMSE | $R^2$ | RMSE | $R^2$ | RMSE | $R^2$ | RMSE | $R^2$ | RMSE |
| Equation (1) | $T_5$ | 0.74 | 1.25 | 0.79 | 1.34 | 0.76 | 1.14 | 0.77 | 1.53 | 0.74 | 2.01 |
| | $T_{10}$ | 0.74 | 1.25 | 0.77 | 1.35 | 0.72 | 1.21 | 0.76 | 1.49 | 0.71 | 2.03 |
| | $T_{15}$ | 0.74 | 1.22 | 0.76 | 1.32 | 0.68 | 1.23 | 0.73 | 1.48 | 0.67 | 2.08 |
| | $LST_{an}$ | 0.64 | 1.50 | 0.73 | 1.54 | 0.79 | 1.13 | 0.74 | 1.48 | 0.71 | 1.97 |
| | $LST_{nightav}$ | 0.65 | 1.46 | 0.73 | 1.54 | 0.78 | 1.17 | 0.72 | 1.55 | 0.70 | 2.03 |
| | $LST_{tn}$ | 0.63 | 1.48 | 0.69 | 1.63 | 0.76 | 1.26 | 0.67 | 1.68 | 0.67 | 2.16 |
| | $LST_{tav}$ | 0.57 | 1.56 | 0.68 | 1.73 | 0.70 | 1.36 | 0.64 | 2.01 | 0.67 | 2.20 |
| | $LST_{av}$ | 0.56 | 1.61 | 0.67 | 1.76 | 0.72 | 1.34 | 0.65 | 2.00 | 0.67 | 2.19 |
| | $LST_{aav}$ | 0.53 | 1.70 | 0.64 | 1.85 | 0.71 | 1.36 | 0.62 | 2.03 | 0.65 | 2.22 |
| | $LST_{td}$ | 0.44 | 1.83 | 0.59 | 1.98 | 0.55 | 1.62 | 0.49 | 2.03 | 0.56 | 2.53 |
| | $LST_{dayav}$ | 0.41 | 1.90 | 0.54 | 2.08 | 0.58 | 1.63 | 0.48 | 2.09 | 0.53 | 2.57 |
| | $LST_{ad}$ | 0.35 | 2.02 | 0.44 | 2.25 | 0.54 | 1.68 | 0.41 | 2.19 | 0.47 | 2.65 |
| Equation (2) | $T_5$ | 0.74 | 1.24 | 0.80 | 1.32 | 0.78 | 1.05 | 0.81 | 1.40 | 0.77 | 1.88 |
| | $T_{10}$ | 0.75 | 1.24 | 0.79 | 1.33 | 0.74 | 1.12 | 0.79 | 1.37 | 0.75 | 1.90 |
| | $T_{15}$ | 0.75 | 1.21 | 0.77 | 1.30 | 0.70 | 1.15 | 0.75 | 1.38 | 0.70 | 1.96 |
| | $LST_{an}$ | 0.62 | 1.51 | 0.71 | 1.56 | 0.79 | 1.09 | 0.75 | 1.42 | 0.74 | 1.85 |
| | $LST_{nightav}$ | 0.63 | 1.46 | 0.72 | 1.54 | 0.80 | 1.10 | 0.74 | 1.47 | 0.74 | 1.89 |
| | $LST_{tn}$ | 0.62 | 1.46 | 0.70 | 1.58 | 0.78 | 1.17 | 0.71 | 1.57 | 0.71 | 1.98 |
| | $LST_{tav}$ | 0.56 | 1.59 | 0.69 | 1.70 | 0.72 | 1.31 | 0.66 | 1.96 | 0.69 | 2.08 |
| | $LST_{av}$ | 0.55 | 1.63 | 0.67 | 1.74 | 0.73 | 1.29 | 0.66 | 1.95 | 0.69 | 2.08 |
| | $LST_{aav}$ | 0.53 | 1.69 | 0.64 | 1.81 | 0.72 | 1.31 | 0.64 | 1.99 | 0.67 | 2.12 |
| | $LST_{td}$ | 0.43 | 1.84 | 0.60 | 1.94 | 0.56 | 1.57 | 0.51 | 1.98 | 0.58 | 2.42 |
| | $LST_{dayav}$ | 0.42 | 1.89 | 0.56 | 2.03 | 0.59 | 1.57 | 0.51 | 2.03 | 0.56 | 2.47 |
| | $LST_{ad}$ | 0.38 | 1.98 | 0.48 | 2.19 | 0.56 | 1.62 | 0.45 | 2.13 | 0.51 | 2.56 |

$R^2$ is the coefficient of determination; and RMSE ($\mu mol\ CO_2\ m^{-2}\ s^{-1}$) is the root mean square error. $T_5$, $T_{10}$, and $T_{15}$ is soil temperature (°C) at a depth of 5, 10, and 15 cm, respectively. $LST_{td}$ and $LST_{tn}$ is the daytime and nighttime land surface temperature observed by MODIS onboard Terra satellites, respectively. $LST_{tav}$ is the mean of $LST_{td}$ and $LST_{tn}$. $LST_{ad}$ and $LST_{an}$ is the daytime and nighttime land surface temperature observed by MODIS onboard Aqua satellites, respectively. $LST_{aav}$ is the mean of $LST_{ad}$ and $LST_{an}$. $LST_{av}$ is the mean of $LST_{td}$, $LST_{tn}$, $LST_{ad}$, and $LST_{an}$. $LST_{dayav}$ is the mean of $LST_{ad}$ and $LST_{td}$, $LST_{nightav}$ is the mean of $LST_{an}$ and $LST_{tn}$. The correlations were all significant at the 0.01 level ($n = 58$).

### 3.3. Correlations between $R_s$ and VIs

The correlations between the $R_s$ and the three VI values were all significant at the 0.01 level for each site. The exponential and linear functions performed comparably in describing the dependency of $R_s$ on the VI values for the five study sites (Table 6). Furthermore, NDVI consistently exhibited a better correlation (with the highest $R^2$ and the lowest RMSE) with $R_s$ than the other two VI values at all sites. Therefore, in the following analysis, we selected NDVI to represent the $R_s$ response to GPP at the seasonal time scale.

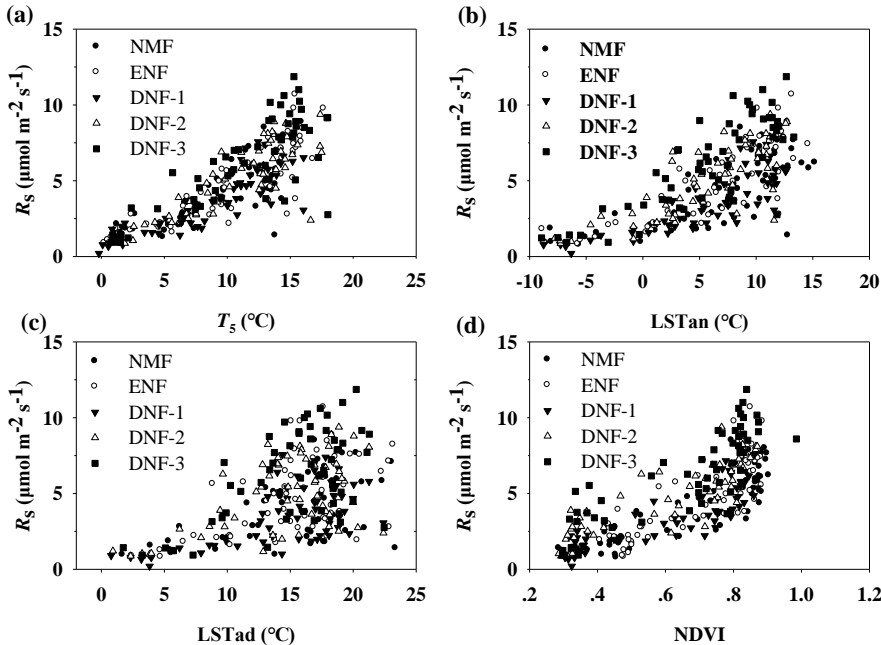

**Figure 2.** Correlations between soil respiration ($R_s$, μmol $CO_2$ m$^{-2}$ s$^{-1}$) and (**a**) soil temperature at the 5 cm depth ($T_5$, °C), (**b**) nighttime land surface temperature from Aqua MODIS (LST$_{an}$, °C), (**c**) daytime land surface temperature from Aqua MODIS (LST$_{ad}$, °C), and (**d**) normalized difference vegetation index (NDVI) during the measurement.

**Table 6.** Results of statistical analysis relating soil respiration to VIs at the five sites for all measured data during measurement [a].

| Model | VI | NMF | | ENF | | DNF-1 | | DNF-2 | | DNF-3 | |
|---|---|---|---|---|---|---|---|---|---|---|---|
| | | $R^2$ | RMSE | $R^2$ | RMSE | $R^2$ | RMSE | $R^2$ | RMSE | $R^2$ | RMSE |
| | NDVI | 0.76 | 1.17 | 0.68 | 1.46 | 0.72 | 0.98 | 0.74 | 1.17 | 0.71 | 1.65 |
| Equation (3) | EVI | 0.65 | 1.44 | 0.63 | 1.70 | 0.65 | 1.37 | 0.63 | 1.65 | 0.61 | 2.03 |
| | CI$_{green edge}$ | 0.66 | 1.49 | 0.54 | 1.76 | 0.57 | 1.63 | 0.60 | 1.71 | 0.60 | 2.11 |
| | NDVI | 0.73 | 1.18 | 0.66 | 1.48 | 0.72 | 1.02 | 0.76 | 1.17 | 0.71 | 1.55 |
| Equation (4) | EVI | 0.66 | 1.32 | 0.60 | 1.59 | 0.65 | 1.13 | 0.64 | 1.42 | 0.63 | 1.77 |
| | CI$_{green edge}$ | 0.70 | 1.23 | 0.60 | 1.60 | 0.66 | 1.12 | 0.69 | 1.33 | 0.71 | 1.55 |

[a] VIs are vegetation indices. NDVI is the normalized difference vegetation index, EVI is the enhanced vegetation index, CI$_{green edge}$ is the green chlorophyll index. $R^2$ is the coefficient of determination, and RMSE (μmol $CO_2$ m$^{-2}$ s$^{-1}$) is the root mean square error. The correlations were all significant at the 0.01 level ($n = 58$).

### 3.4. Combined Correlations between $R_s$ and $T_s$ (or LST) and NDVI

When $T_5$ (or LST$_{an}$) and NDVI were integrated into one of the six two-variable models (Equations (5)–(10)), the results showed that each of the fitted equations could be used to precisely predict $R_s$ from $T_5$ (or LST$_{an}$) and NDVI variables (Table 7). Furthermore, in comparison with the one-dimensional equation (Equations (1)–(4); Tables 5 and 6), two-variable models (Table 7) were better for all five sites based on the value of RMSE and AIC. According to the model performance indicators ($R^2$, RMSE, and AIC), the performances of the fitted $T_5$–NDVI models were very similar to the fitted LST$_{an}$–NDVI model, due to the fact that with the independent t test for the $R^2$, RMSE, and AIC values between the $R_s$ to $T_5$ and NDVI model and the $R_s$ to LST and NDVI model, except for the $R^2$, demonstrated a significant difference. However, it was not the case in RMSE and AIC.

### 3.5. Modeled Soil Respiration Validation

The results obtained from the leave one out cross-validation are shown in Table 8. In contrast to the cross-validation statistical result of the one-dimensional equation, the $R^2$ and EF of the two-dimensional

equations increased, and RMSE decreased. This further confirmed our conclusion that the application of the two-dimensional equations of $T_5$–NDVI (or $LST_{an}$–NDVI) is better than the one-dimensional equations of $T_5$ (or $LST_{an}$) in predicting $R_s$ at a seasonal scale. Furthermore, the cross-validated statistics of the models driven by $LST_{an}$ or $LST_{an}$–NDVI were slightly lower than those of the models of the in situ measured $T_5$ or $T_5$–NDVI.

The modeled $R_s$ closely resembled the seasonal patterns of the measured $R_s$ (Figure 3). $R_s$ increased quickly after the start of the growing season and maximized in the summer months and then underwent an evident decrease since autumn. An obvious overestimation that occurred on 5 July 2009 at the five forest sites reduced the evaluation accuracy of the cross-validation because of summer drought. When 2009 was excluded from the model validation, the $R^2$ and EF of the cross-validation increased and RMSE decreased compared with that including all measured data for one of the five sites (Table S2). Moreover, underestimation was observed after raining or the middle growing period.

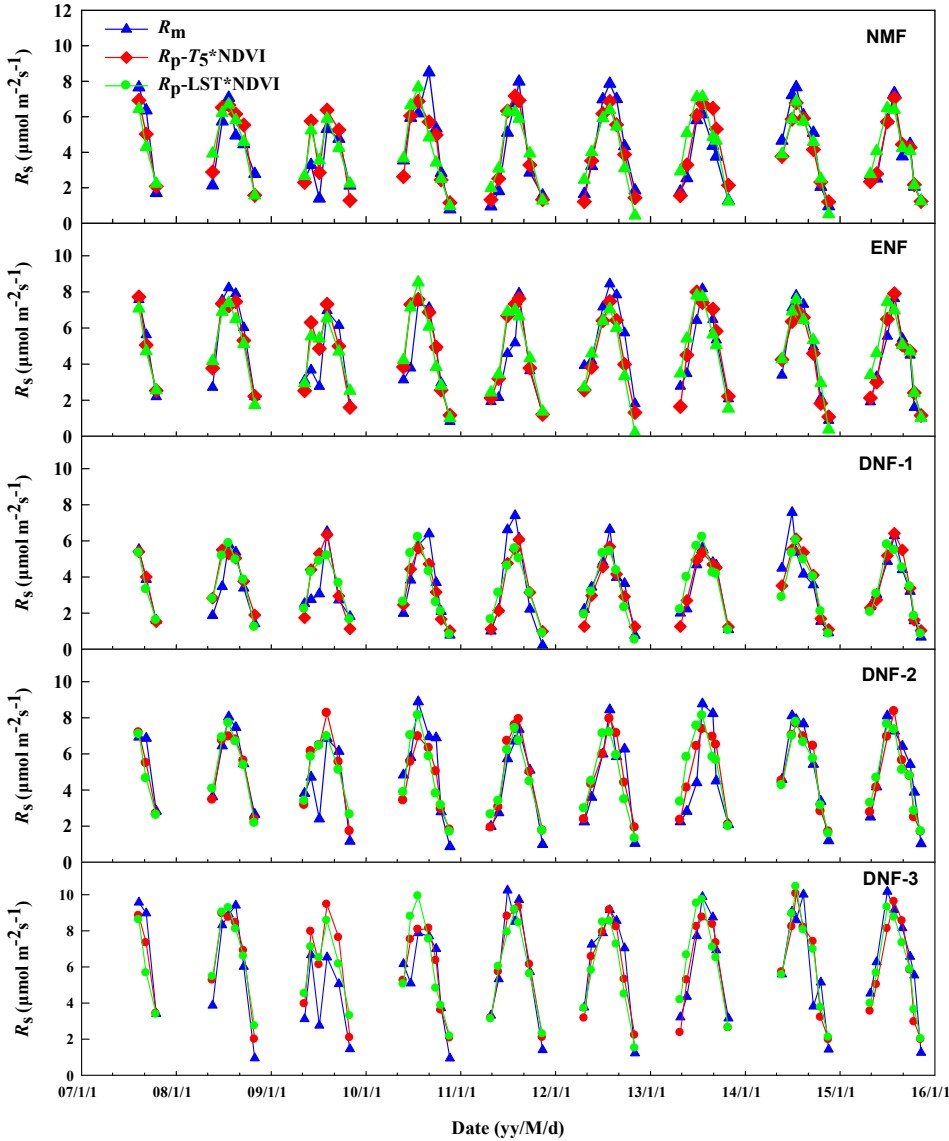

**Figure 3.** Seasonal variations of soil respiration measured ($R_m$) and predicted ($R_p$) by model five based on in situ soil temperature and NDVI ($R_p$–$T_5$*NDVI) or nighttime LST and NDVI ($R_p$–$LST_{an}$*NDVI) for the five study sites.

**Table 7.** Fitting statistics for the respiration models at the five sites for all measured data from during the measurement.

| Equation | NMF | | | ENF | | | DNF-1 | | | DNF-2 | | | DNF-3 | | |
|---|---|---|---|---|---|---|---|---|---|---|---|---|---|---|---|
| | $R^2$ | RMSE | AIC | $R^2$ | RMSE | AIC | $R^2$ | RMSE | AIC | $R^2$ | RMSE | AIC | $R^2$ | RMSE | AIC |
| Soil temperature at 5 cm depth | | | | | | | | | | | | | | | |
| $R_s = a \times e^{b \times T}$ | 0.74 | 1.25 | 29.59 | 0.79 | 1.34 | 37.67 | 0.76 | 1.14 | 19.19 | 0.77 | 1.53 | 53.68 | 0.74 | 2.01 | 85.13 |
| $R_s = R_{ref} \times e^{(b(1/56.02-1/(T+46.02)))}$ | 0.74 | 1.24 | 28.89 | 0.80 | 1.32 | 36.27 | 0.78 | 1.05 | 10.10 | 0.81 | 1.40 | 43.10 | 0.77 | 1.88 | 77.06 |
| Soil temperature at 5 cm depth and NDVI | | | | | | | | | | | | | | | |
| $R_s = a + b \times T \times VI$ | 0.82 | 0.96 | −1.21 | 0.79 | 1.16 | 20.74 | 0.80 | 0.85 | −14.18 | 0.79 | 1.10 | 15.08 | 0.77 | 1.51 | 51.47 |
| $R_s = a + b \times T + c \times VI$ | 0.80 | 1.02 | 8.21 | 0.77 | 1.21 | 27.75 | 0.78 | 0.91 | −5.40 | 0.80 | 1.08 | 14.39 | 0.76 | 1.52 | 54.79 |
| $R_s = a \times e^{(b \times T + c \times VI)}$ | 0.84 | 1.10 | 17.24 | 0.84 | 1.17 | 23.81 | 0.79 | 0.90 | −6.01 | 0.81 | 1.18 | 25.67 | 0.78 | 1.57 | 58.38 |
| $R_s = a \times e^{b \times T} \times VI^c$ | 0.85 | 1.10 | 17.30 | 0.84 | 1.17 | 24.03 | 0.79 | 0.91 | −5.15 | 0.82 | 1.18 | 24.87 | 0.79 | 1.56 | 57.64 |
| $R_s = R_{ref} \times e^{((b(1/56.02-1/(T+46.02)))+c \times VI)}$ | 0.84 | 1.10 | 16.71 | 0.85 | 1.16 | 23.22 | 0.80 | 0.88 | −8.32 | 0.84 | 1.16 | 22.86 | 0.81 | 1.55 | 56.89 |
| $R_s = R_{ref} \times e^{((b(1/56.02-1/(T+46.02)))} \times VI^c$ | 0.85 | 0.96 | 1.36 | 0.85 | 1.16 | 23.55 | 0.80 | 0.90 | −6.63 | 0.84 | 1.16 | 23.24 | 0.81 | 1.55 | 57.06 |
| Nighttime LST from Aqua MODIS | | | | | | | | | | | | | | | |
| $R_s = a \times e^{b \times LST}$ | 0.64 | 1.50 | 50.92 | 0.73 | 1.54 | 54.19 | 0.79 | 1.13 | 18.00 | 0.74 | 1.48 | 49.17 | 0.71 | 1.97 | 82.56 |
| $R_s = R_{ref} \times e^{(b(1/56.02-1/(LST+46.02)))}$ | 0.62 | 1.51 | 52.13 | 0.71 | 1.56 | 55.94 | 0.79 | 1.09 | 13.86 | 0.75 | 1.42 | 44.28 | 0.74 | 1.85 | 75.11 |
| Nighttime LST from Aqua MODIS and NDVI | | | | | | | | | | | | | | | |
| $R_s = a + b \times LST \times VI$ | 0.71 | 1.22 | 26.91 | 0.71 | 1.37 | 40.37 | 0.74 | 0.97 | 0.71 | 0.71 | 1.28 | 32.94 | 0.70 | 1.66 | 63.04 |
| $R_s = a + b \times LST + c \times VI$ | 0.75 | 1.12 | 18.96 | 0.72 | 1.34 | 39.70 | 0.75 | 0.97 | 2.58 | 0.77 | 1.15 | 22.21 | 0.74 | 1.57 | 58.03 |
| $R_s = a \times e^{(b \times LST + c \times VI)}$ | 0.81 | 1.11 | 18.56 | 0.81 | 1.31 | 37.76 | 0.81 | 0.98 | 3.99 | 0.79 | 1.20 | 27.43 | 0.77 | 1.64 | 63.53 |
| $R_s = a \times e^{b \times LST} \times VI^c$ | 0.81 | 1.11 | 17.80 | 0.80 | 1.32 | 38.26 | 0.81 | 1.00 | 5.54 | 0.79 | 1.20 | 27.28 | 0.77 | 1.61 | 61.37 |
| $R_s = R_{ref} \times e^{((b(1/56.02-1/(LST+46.02)))+c \times VI)}$ | 0.81 | 1.11 | 17.81 | 0.81 | 1.30 | 36.65 | 0.82 | 0.95 | −0.38 | 0.81 | 1.17 | 24.70 | 0.79 | 1.60 | 60.20 |
| $R_s = R_{ref} \times e^{((b(1/56.02-1/(LST+46.02)))} \times VI^c$ | 0.82 | 1.11 | 17.60 | 0.80 | 1.31 | 37.55 | 0.82 | 0.97 | 2.56 | 0.81 | 1.19 | 26.08 | 0.79 | 1.59 | 59.75 |

$R^2$ is the coefficient of determination, RMSE ($\mu$mol $CO_2$ m$^{-2}$ s$^{-1}$) is the root mean square error, and AIC is a version of Akaike's information criterion. $T_5$ is soil temperature at the 5-cm depth. $LST_{an}$ is the nighttime land surface temperature from Aqua MODIS. VI is the normalized difference vegetation index (NDVI). All relationships were statistically significant at $p < 0.01$ ($n = 58$).

**Table 8.** The leave one out cross-validation statistics for the respiration models of the five sites during the measurement.

| Equation | NMF $R^2$ | RMSE | EF | ENF $R^2$ | RMSE | EF | DNF-1 $R^2$ | RMSE | EF | DNF-2 $R^2$ | RMSE | EF | DNF-3 $R^2$ | RMSE | EF |
|---|---|---|---|---|---|---|---|---|---|---|---|---|---|---|---|
| **Soil temperature at 5 cm depth** | | | | | | | | | | | | | | | |
| $R_s = a \times e^{b \times T}$ | 0.76 | 1.36 | 0.48 | 0.79 | 1.25 | 0.31 | 0.78 | 1.19 | 0.27 | 0.77 | 1.50 | 0.21 | 0.74 | 1.88 | −0.32 |
| $R_s = R_{ref} \times e^{(b(1/56.02-1/(T+46.02)))}$ | 0.76 | 1.37 | 0.49 | 0.79 | 1.25 | 0.33 | 0.80 | 1.10 | 0.45 | 0.78 | 1.37 | 0.36 | 0.76 | 1.73 | −0.05 |
| **Soil temperature at 5 cm depth and VI** | | | | | | | | | | | | | | | |
| $R_s = a = b \times T \times VI$ | 0.81 | 1.18 | 0.63 | 0.87 | 1.01 | 0.63 | 0.84 | 0.87 | 0.70 | 0.84 | 1.09 | 0.66 | 0.84 | 1.40 | 0.47 |
| $R_s = a = b \times T + c \times VI$ | 0.79 | 1.26 | 0.59 | 0.81 | 1.11 | 0.53 | 0.80 | 1.00 | 0.60 | 0.81 | 1.11 | 0.64 | 0.80 | 1.58 | 0.31 |
| $R_s = a \times e^{(b \times T + c \times VI)}$ | 0.81 | 1.20 | 0.60 | 0.83 | 1.10 | 0.56 | 0.80 | 1.05 | 0.53 | 0.82 | 1.25 | 0.48 | 0.78 | 1.71 | −0.06 |
| $R_s = a \times e^{b \times T} \times VI^c$ | 0.81 | 1.20 | 0.60 | 0.87 | 1.03 | 0.61 | 0.83 | 0.97 | 0.59 | 0.81 | 1.24 | 0.50 | 0.78 | 1.66 | 0.06 |
| $R_s = R_{ref} \times e^{((b(1/56.02-1/(T+46.02)))+c \times VI)}$ | 0.81 | 1.21 | 0.61 | 0.85 | 1.07 | 0.58 | 0.83 | 0.95 | 0.62 | 0.82 | 1.21 | 0.53 | 0.79 | 1.64 | 0.06 |
| $R_s = R_{ref} \times e^{((b(1/56.02-1/(T+46.02)))} \times VI^c$ | 0.81 | 1.21 | 0.61 | 0.86 | 1.04 | 0.61 | 0.83 | 0.95 | 0.62 | 0.82 | 1.21 | 0.53 | 0.79 | 1.62 | 0.12 |
| **Nighttime LST from Aqua MODIS** | | | | | | | | | | | | | | | |
| $R_s = a \times e^{b \times LST}$ | 0.67 | 1.58 | 0.35 | 0.66 | 1.49 | 0.22 | 0.71 | 1.15 | 0.46 | 0.69 | 1.41 | 0.44 | 0.67 | 1.87 | 0.00 |
| $R_s = R_{ref} \times e^{(b(1/56.02-1/(LST+46.02)))}$ | 0.66 | 1.60 | 0.37 | 0.64 | 1.50 | 0.26 | 0.72 | 1.09 | 0.54 | 0.71 | 1.37 | 0.48 | 0.71 | 1.77 | 0.15 |
| **Nighttime LST from Aqua MODIS and VI** | | | | | | | | | | | | | | | |
| $R_s = a = b \times LST \times VI$ | 0.73 | 1.40 | 0.49 | 0.76 | 1.24 | 0.49 | 0.77 | 1.00 | 0.62 | 0.75 | 1.19 | 0.59 | 0.78 | 1.58 | 0.34 |
| $R_s = a = b \times LST = c \times VI$ | 0.74 | 1.37 | 0.53 | 0.79 | 1.16 | 0.55 | 0.76 | 1.00 | 0.61 | 0.77 | 1.55 | 0.38 | 0.80 | 1.57 | 0.41 |
| $R_s = a \times e^{(b \times LST + c \times VI)}$ | 0.76 | 1.31 | 0.54 | 0.82 | 1.20 | 0.50 | 0.76 | 1.03 | 0.58 | 0.77 | 1.16 | 0.61 | 0.76 | 1.68 | 0.23 |
| $R_s = a \times e^{b \times LST} \times VI^c$ | 0.76 | 1.33 | 0.53 | 0.81 | 1.13 | 0.58 | 0.76 | 1.04 | 0.58 | 0.77 | 1.16 | 0.61 | 0.77 | 1.63 | 0.31 |
| $R_s = R_{ref} \times e^{((b(1/56.02-1/(LST+46.02)))+c \times VI)}$ | 0.77 | 1.30 | 0.55 | 0.81 | 1.15 | 0.57 | 0.78 | 0.98 | 0.63 | 0.78 | 1.12 | 0.64 | 0.76 | 1.65 | 0.25 |
| $R_s = R_{ref} \times e^{((b(1/56.02-1/(LST+46.02)))} \times VI^c$ | 0.76 | 1.32 | 0.55 | 0.81 | 1.15 | 0.57 | 0.77 | 0.99 | 0.62 | 0.78 | 1.14 | 0.63 | 0.77 | 1.60 | 0.34 |

$R^2$ is the coefficient of determination, RMSE is the root mean square error, and EF is the modeling efficiency. The correlations were all significant at the 0.01 level ($n = 58$).

## 4. Discussion

### 4.1. The Impact of Temperature on $R_s$

The LST data from MODIS products can potentially be used as a measure of temperature [17]. At our study site, MODIS Terra and Aqua LST values were all significantly correlated with the observed $T_s$ (Table 4). When comparing the correlation between nighttime LST values and in situ observed $T_s$ values with that of the correlation of daytime LST and in situ observed $T_s$, the better nighttime correlation could be attributed to both the absence of a solar radiation effect on the thermal infrared signal at night [26] and the influence of vegetation during daytime [27]. During the daytime, dense vegetation may increase the conversion of solar incident energy into latent heat, and thus cool the surface through evapotranspiration. During the nighttime, vegetation exerts a negligible effect on the correlation between surface air temperature and nighttime LST [19,27].

In our study, we found that the MODIS LST data could be used to establish models estimating $R_s$, as an alternative to $T_s$. Our results also show that nighttime LST data were usually better correlated with $R_s$ than daytime LST, as indicated by the performance of the exponential functions and the Arrhenius-type functions (Table 5). It was concluded that nighttime LST was the optimal predictor for estimating $R_s$. This might be attributed to the nighttime LST values, indicating the baseline temperature regulates plant phenology [17].

Our result is consistent with previous studies [19,20,27]. For example, in a Canadian boreal black spruce stand, Wu et al. [20] reported that nighttime LST showed a greater potential in explaining variations in $R_s$ than daytime LST, referring to the fact that nighttime LST is more resistant to various residual noise components. Huang et al. [19] suggests that an accurate estimation of $R_s$ could be inferred with Terra MODIS LST using either nighttime LST or the mean of daytime and nighttime LST as the independent variable in regression equations.

### 4.2. Vegetation Index as a Driver of Rs

We identified that $R_s$ was correlated with three kinds of VIs (i.e., NDVI, EVI, and $CI_{green\ edge}$). Others observed the same phenomenon [28,29]. This result suggests that the spectral vegetation indexes from remote sensing can be used in the prediction model of $R_s$. There was a consistently stronger correlation between $R_s$ and NDVI than the correlation between $R_s$ and EVI or $CI_{green\ edge}$ for the five forest sites (Table 6), which was not consistent with others. For example, at a maize and a winter wheat field, Huang et al. [28] reported that EVI or $CI_{red\ edge}$ consistently exhibited a better correlation with $R_s$ than NDVI, which was attributed to NDVI showing less of a seasonal variation than EVI and $CI_{red\ edge}$, particularly when the green leaf area index (GLAI) was greater than 3. Huete et al. [22] also reported that NDVI tends to saturate at high vegetation densities, and is highly sensitive to differences in background reflectance. Conversely, EVI and $CI_{red\ edge}$ improved the canopy background correction and are more sensitive than NDVI to variation in dense vegetation. In our study, the vegetation types of the five study sites are all cold temperate coniferous forests, and the in situ measured monthly LAI ranged from 1.85 to 4.22, 3.08 to 4.77, 1.27 to 2.96, 0.80 to 2.27, and 0.74 to 2.84 from April 2015 to October 2015 at the NMF, ENF, DNF-1, DNF-2, and DNF-3 sites, respectively (Figure 4). Compared with the other reports, our study sites illustrated a sparse vegetation region, and a lower vegetation index. Therefore, NDVI consistently exhibited a better correlation with $R_s$ than the other two VI values. Vegetation indexes at the ENF site are not as well correlated with $R_s$ than that at other sites. This may be attributed to the fact that the ENF site is an evergreen needle leaf forest and the coefficient of variation of its vegetation coverage within a year is the least one among the five sites.

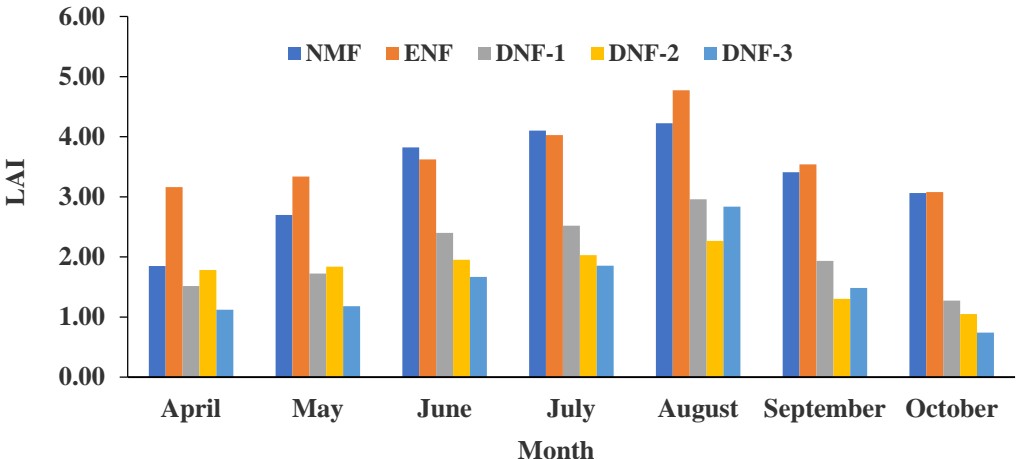

**Figure 4.** The seasonal variations of leaf area index (LAI) from April to October 2015.

### 4.3. Spatial Scale of the Data

The spatial scales of the data for analysis in our study are different. $R_s$, $T_s$, and $W_s$ measurements at each site were carried out in an area of approximately 400 m$^2$, and their values were averaged. However, each pixel of the MODIS 8-day surface reflectance and 8-day LST products represents an area of 500 m × 500 m and 1000 m × 1000 m, respectively. Therefore, the MODIS products (i.e., VI and LST values) are not necessarily consistent with in situ measured data (i.e., $R_s$, $T_s$, and $W_s$) in the spatial scale. Despite that, we found that the MODIS LST and the measured $T_s$ showed a consistent seasonal variation pattern (Figure 1). In addition, a Pearson correlation analysis showed that the MODIS LST values and the measured $T_s$ (i.e., $T_5$, $T_{10}$, and $T_{15}$) were all significantly correlated at the 0.01 level for the five forest sites (Table 4). Huang et al. [3] observed the similar pattern. In our study site in a sub-alpine meadow, it is feasible to predict both $R_s$ with MODIS products and the in situ measured soil temperature [30]. Recently, MODIS data have also been confirmed to estimate ecosystem respiration on the global scale [31]. Therefore, MODIS LST may be identified as a proxy indicator of $T_s$ to estimate $R_s$.

### 4.4. Limitation of the Study

The models driven by remote sensing data (i.e., $LST_{an}$ and NDVI) performed well in $R_s$ estimation at the current five forest sites; however, several limitations are listed below:

We focused our study on the growing season of five temperate coniferous forest sites; therefore, the model's performance in the non-growing season still needs to be evaluated. The previous study also reported that the factors influencing $R_s$ in different phenological phases may be different. For example, Huang et al. [3] found the models driven by mean LST and root zone soil moisture could explain most of the non-growing season's variations in $R_s$ at a deciduous forest site. The models including the mean LST, root zone soil moisture, and EVI exhibited a high accuracy for $R_s$ estimation in early and late-growing periods. However, in the mid-growing period, the model entirely dependent on mean LST, root zone soil moisture, and EVI may exhibit a lower explanation capacity for seasonal variation of $R_s$ than the model driven by in situ measured $T_s$ at the 4 cm depth, $W_s$ at the 10 cm depth, and gross primary productions.

Because the $W_s$ factor was not explicitly included in the prediction models of $R_s$, the satellite-driven model may provide a relatively poor $R_s$ estimation under severe drought and raining pulse conditions, as we observed. Throughout the nine-year study period, $W_s$ was below 50% of WHC only in the measurement on July 2009, where the corresponding $R_s$ was also uniformly lower than that in the same period in other years at the five study sites. An obvious overestimation of $R_s$ was also found on 5 July 2009 (Figure 3). The overestimation may be attributed to the model being based only on $LST_{an}$ and NDVI and not including the effect of $W_s$ on $R_s$. In order to further explore the impact of $W_s$ on the performances of models only based on $T_5$ or $LST_{an}$, a regression analysis was conducted using

logarithmic and parabolic models. The result showed a mean $W_s$ from May to October (MAW) or mean $W_s$ from May to July (MTW) was significantly correlated with the $R^2$ of the exponential model of $T_5$ or $LST_{an}$ to $R_s$ (Table S3) except the DNF-2 and DNF-3 sites. The correlation with MTW was better than with MAW in most cases, indicating that the performance of the $R_s$ model based on $T_5$ or $LST_{an}$ was influenced by $W_s$. Wu et al. [20] also reported that the $R_s$ model based on the data of MODIS LST and NDVI is affected by the soil water amount.

Remotely sensed LST data lack observations for cloud-covered areas [32]. The soil respiration measurement includes both sunny and cloudy days. Moreover, in our study, VI and LST values corresponding to the $R_s$ measurement days were from the two consecutive 8-day composites by linear interpretation. Consequently, information extraction errors from remote sensing data may introduce errors into $R_s$ prediction.

The value of $R_s$ is also influenced by soil texture, substrate quantity, and quality [33]. These factors were not incorporated into our model, however in future studies these factors may improve the accuracy of the $R_s$ models and should be investigated further.

## 5. Conclusions

We investigated the feasibility of estimating $R_s$ using solely MODIS product data on five cold temperate coniferous forest sites in the eastern Loess Plateau, China. The results showed that the accuracy of the model based on the observed surface soil temperatures was not significantly different with that of the model based on MODIS-derived nighttime LST values. However, the model using MODIS-derived daytime LST values was significantly different, indicating that nighttime land surface temperatures were the optimum LST for estimating $R_s$. Between the selected three VI values, NDVI consistently exhibited a better correlation with $R_s$, compared to EVI and $CI_{green\ edge}$. Adding NDVI into the model considering only $T_s$ or nighttime LST significantly improved the simulation accuracy of $R_s$. The models driven by $LST_{an}$ and NDVI demonstrated a similar performance to the models considering $T_5$ and NDVI based on the test of the AIC, $R^2$, and RMSE values. Our findings demonstrate that models based entirely on remote sensing data have the potential to predict $R_s$ in the cold temperate coniferous forest sites. Our previous study and other researches at different sites and vegetation types also have confirmed the feasibility of estimating $R_s$ using solely MODIS product data. The present study provides valuable information for the large-scale estimation of $R_s$ in cold temperate deciduous forest ecosystems. It is possible that the use of MODIS data for soil respiration estimation will provide a great convenient way for forest carbon budget calculation at larger scales.

**Supplementary Materials:** The following are available online at http://www.mdpi.com/1999-4907/11/2/131/s1, Figure S1: Scatter plots between soil respiration ($R_s$) and soil water content ($W_s$) at 0–10 cm depth, Table S1: The statistical analysis of one-way ANOVA based on $R^2$ and RMSE of the model driven by in situ $T_s$ or remote sensing surface temperature. Table S2: The leave one out cross validation statistics for the respiration models of the five sites when the site-year 2009 was exclude from model validation due to severe drought, Table S3: The statistical analysis of regression functions between $R^2$ of the exponential model of $R_s$ to $T_5$ or $LST_{an}$ and mean soil water content from May to October (MAW) or mean soil water content from May to July every year (MTW).

**Author Contributions:** J.Y. designed the field experiments and conducted the data analysis and finished the writing of the paper. X.Z., J.L., and H.L. performed the field experiments. G.D. was for partial data analyses. All authors have read and agreed to the published version of the manuscript.

**Funding:** This study was financially supported by the National Natural Science Foundation of China (Grant No. 41201374, 41130528), Higher Education Institution Project of Shanxi Province: Ecological Remediation of Soil Pollution Disciplines Group (Grant No.:20181401).

**Acknowledgments:** The authors thank all postgraduate students for their valuable help in the fieldwork. We would like to thank Editage (www.editage.cn) for English language editing. We also appreciate the two anonymous reviewers for their insightful comments and suggestions to improve the quality of our manuscript.

**Conflicts of Interest:** The authors declare no conflict of interest.

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
