# Peer review of "MODIS-Derived Estimation of Soil Respiration within Five Cold Temperate Coniferous Forest Sites in the Eastern Loess Plateau, China"

_forests, doi:10.3390/f11020131_

Round 1

Reviewer 2 Report

Specific comments on MODIS-derived estimation of soil respiration within five cold

temperate coniferous forest sites in the eastern Loess Plateau, China”.

This manuscript adds further weight to the existing body of literature describing the ability of remote sensing data products to predict soil respiration. In the abstract the authors comment that the performance of the model in other vegetation types needs further study. However, it seems the authors have already published a similar study in sub alpine meadows in Northern China (Sustainability 11, 3214; doi:10.3390/su11123274). This current manuscript would benefit from some discussion on the state of the art of this area of research, recognising that the approach has been used elsewhere. For example, Jinlong Ai in the Journal of geophysical research: Biogeosciences vol 123, issue 2 (Doi.org/10.1002/2017JG004107) ) describes results from 171 sites distributed around the globe.

The authors should also add some commentary about this area of research can be improved and how the models can be brought into practice to improve management and policy in the land management debate.

Author Response

Dear Sir/Madam:

Thank very much for your consideration to publish our paper.

According to your suggestion, we added two more references. One is a paper we have published in Sustainability (11, 3214; doi: 10.3390/su11123274), which was about describing the ability of remote sensing data products to predict soil respiration in sub alpine meadows in Northern China. The other is from the Journal of geophysical research: Biogeosciences (vol. 123, issue 2, Doi.org/10.1002/ 2017JG004107) which developed a remote sensing model estimating ecosystem respiration across most biomes. We also added some detailed information about the forest character in the site section. In the discussion session, we also have some suggestions for forest management.

Sincerely,

Junxia Yan

PhD, associated professor

Institute of Loess Plateau, Shanxi University

Environmental Building

92 Wucheng Road

Taiyuan, Shanxi Province 030006, P. R. China

Tel: +86 351-7010700/+86 18634348212
